# Trial-by-trial learning of successor representations in human behavior

**Ari E. Kahn**[1]*, **Dani S. Bassett**[2], **Nathaniel D. Daw**[1,3]

**1** Princeton Neuroscience Institute, Princeton University, Princeton, New Jersey, United States of America, **2** Department of Bioengineering, University of Pennsylvania, Philadelphia, Pennsylvania, United States of America, **3** Department of Psychology, Princeton University, Princeton, New Jersey, United States of America

\* arikahn@arizona.edu

## Abstract

Decisions in humans and other organisms depend, in part, on learning and using models that capture the statistical structure of the world, including the long-run expected outcomes of our actions. One prominent approach to forecasting such long-run outcomes is the successor representation (SR), which predicts future states aggregated over multiple timesteps. Although much behavioral and neural evidence suggests that people and animals use such a representation, it remains unknown how they acquire it. It has frequently been assumed to be learned by temporal difference bootstrapping (SR-TD(0)), but this assumption has largely not been empirically tested or compared to alternatives including eligibility traces (SR-TD($\lambda > 0$)). Here we address this gap by leveraging trial-by-trial reaction times in graph sequence learning tasks, which are favorable for studying learning dynamics because the long horizons in these studies differentiate the transient update dynamics of different learning rules. We examined the behavior of SR-TD($\lambda$) on a probabilistic graph learning task alongside a number of alternatives, and found that behavior was best explained by a hybrid model which learned via SR-TD($\lambda$) alongside an additional predictive model of recency. The relatively large $\lambda$ we estimate indicates a predominant role of eligibility trace mechanisms over the bootstrap-based chaining typically assumed. Our results provide insight into how humans learn predictive representations, and demonstrate that people simultaneously learn the SR alongside lower-order predictions.

## Author summary

Our ability to plan intelligently requires predicting the state of the world multiple steps into the future. Enumerating future outcomes step-by-step, however, is slow and costly. Instead, research has shown that people rely on simplified models of

**Data availability statement:** Data and code that support this study are available at https://osf.io/jupbz/.

**Funding:** This work was supported by the National Institute of Mental Health R01MH136875 (N.D.D.) and R01MH135587 (N.D.D.), part of the CRCNS program, and National Institute of Health R21-MH-106799 (D.S.B.). The funders had no role in study design, data collection and analysis, decision to publish, or preparation of the manuscript.

**Competing interests:** The authors have declared that no competing interests exist.

the world that skip across multiple steps at once. How do we construct these simplified models? One promising idea is the successor representation (SR), which predicts future events via a simple and neurally plausible computation. The SR has been shown to explain a range of behavioral phenomena, but these studies have not identified which among many learning rules the brain uses to build the SR. Plausible mechanisms for learning associations over delays (called bootstrapping and eligibility traces) both converge to identical simplified world models, and thus existing studies on the SR, which focus on well trained behavior, are unable to distinguish between them. Here, we answer this question by examining behavior on a graph learning task, where stimulus-by-stimulus reaction times have been shown to reflect predictions over long temporal horizons. Through both model fitting and model-agnostic comparisons, we find that behavior is best explained by a learning rule heavily dependent on eligibility traces, in contrast to previous work which generally assumed an (untested) bootstrapping update rule.

## Introduction

Decisions in humans and other organisms depend, in part, on learning and using models that capture the statistical structure of the world, including the long-run expected outcomes of our actions. A range of neural and behavioral results [1–8] suggest that the brain uses temporally abstract representations like the successor representation (SR) [9], which predict future events aggregated over multiple steps. However, such long-range predictions are challenging to learn, and it remains unknown how the brain accomplishes this learning.

Although the SR has often been studied via its effects on choices in simple sequential decision tasks [4,10], trial-by-trial learning dynamics are difficult to study in this setting due to short temporal horizons and the indirect observation of predictions by way of choices. One setting in which it may be particularly promising to investigate such learning rules is stimulus-by-stimulus reaction times (RTs) in graph learning tasks, which reflect predictions more directly and over long temporal horizons. In particular, previous work has shown that in such tasks, people exhibit systematic RT biases. Among these is sensitivity to modular structure, which manifests as a 'cross-cluster surprisal effect' where RTs are heightened when transitioning between densely interconnected clusters of nodes [11,12]. RT facilitation when transitioning from one state to another has been interpreted as reflecting learned expectancy, but the cross-cluster effect implies that these expectancies are not captured by simple one-step conditional probabilities (which are matched on cluster transition). Instead, these behavioral biases were shown to be well captured by a maximum-entropy prediction model that balances representational complexity with accuracy [13], and which is mathematically equivalent to the SR. The SR's predictions are higher within than between clusters, because multi-step transitions move between clusters relatively rarely. These results suggest that peoples' learned representations of graph structure (as revealed by predictive facilitation of their RTs) reflect an SR [14].

While these effects occur in block-averaged behavior, and are predicted asymptotically by a well-learned SR, there are a number of distinct hypothesized mechanisms for how the SR might be learned in a trial-by-trial fashion [15]. Such learning is computationally challenging and the mechanisms supporting it are of some interest, because it requires tracking long-run connections between multiple states—much like other cases of credit assignment but over multiple destination states simultaneously.

There are multiple candidate mechanisms for learning rules that support long-run forecasting. Since the first article introducing the SR [9], it has been widely assumed that the SR is directly learned via temporal-difference learning ("SR-TD"), specifically TD(0) bootstrapping [9,16,17]. TD(0) learns long-run predictions by serially chaining single-step "bootstrap" operations, backing up the prediction vector for each newly encountered state to its immediate predecessor and thereby spanning longer gaps via repeated traversals. Additionally, although less explored in this context, TD bootstrapping can be augmented or replaced by additionally maintaining "eligibility traces" over recently encountered states, allowing updates directly to span longer temporal gaps (SR-TD($\lambda > 0$)). A potential role for eligibility traces is intriguing in part because eligibility traces can also operate independently alongside or instead of bootstrapping (e.g., SR-TD(1) itself approximates Monte Carlo learning which can also be accomplished by combining eligibility traces with a simpler, biologically plausible Hebbian update), and because there have been recent suggestions that trace-driven learning might be instantiated in the brain using backward trajectory replay [18–20]. Further, the entropy-regularized prediction model developed by [13] suggests a learning rule that also, at least in expectation, corresponds to Hebbian learning augmented with eligibility traces. Importantly, although trace- and bootstrap-based learning rules converge to the same asymptotic representation of future occupancy, they do so via distinct trial-by-trial dynamics.

In the current study, we empirically test the fits of SR learning algorithms to the trial-by-trial dynamics of RTs in sequential graph learning tasks (Fig 1A) , by analyzing learning in three graphs (Fig 1B) using the data from [12]. If RTs on these tasks reflect long-run expectancy [21], then transient fluctuations in learned SRs due to variation in sequences should co-vary with biases in reaction times, allowing us to characterize the learning rules driving peoples' representations. Additionally, the update methods of these algorithms (in particular, bootstrapping and traces) should themselves have distinct signatures observable in reaction times, by comparing transitions that either do or do not affect either mechanism.

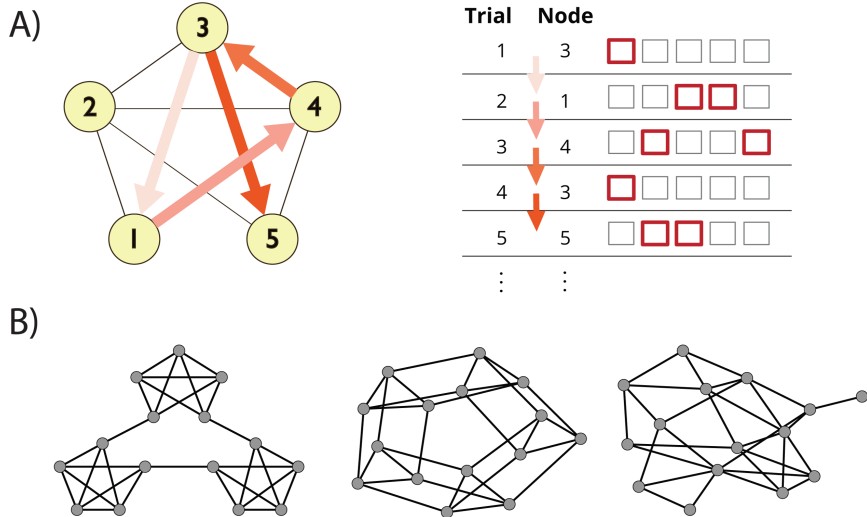

**Fig 1**. **SR prediction on a graph learning task. A)** Schematic of task design. Participants responded to a sequence of 1500 stimuli derived from a random walk on a graph, where each trial required pressing a one- or two-button combination of keys to identify the presented stimulus as quickly as possible. **B)** Construction of stimulus sequences. Walks were generated from either a modular (left), lattice (center), or random graph (right).

## Results

### Successor representation

The SR is a world model that enables planning via cached long-run predictions of future state. That is, it learns to predict $M(s, s') = \sum_{t=0}^{\infty} \gamma^t p(s_t = s' | s_0 = s)$, a temporally discounted future occupancy of state $s'$ given state $s$. In a tabular setting, this can be represented via an $n \times n$ matrix for $n$ discrete states, where $M_{s,s'} = M(s, s')$. By coupling this model with a one-step estimated reward at each state, $R(s)$, the long-run value of a state can be calculated as the dot product $V(s) = M_{s,:}.R$. (One-step estimated rewards can be trivially learned, for example, with a delta rule.)

Candidate learning rules for the SR itself, however, are more varied, notably featuring two distinct mechanisms for bridging temporal gaps: serially chaining one-step outcomes via bootstrapping and broadcasting them via eligibility traces. These two mechanisms, in turn, represent special cases of the SR-TD($\lambda$) algorithm, which interpolates between them (see Methods). While these rules will, in appropriate conditions, converge asymptotically to the same matrix $M$, their transient behavior prior to (and indeed after) convergence is quite different, exhibiting unique biases due to bootstrapping and trace updates. By using RTs to read out internal predictions, we hoped to test which (or what combination) of these learning rules would best reflect recent trial history in subject behavior on a graph learning task.

### Recency learning model best predicts reaction times

What learning model best captures the transient trial-by-trial reaction time biases observed in graph learning tasks? Among those proposed [15], a plausible range of possibilities are parameterized by SR-TD($\lambda$), which learns $M$ via temporal difference learning. It does so by computing a bootstrapped TD error $\delta = I_j + \gamma M_{j,:} - M_{i,:}$ on each trial, and updating previous states based on this error via a trace whose decay is parameterized by $\lambda$. SR-TD($\lambda$) thus encapsulates a family of models, where $\lambda = 0$ only updates the immediately preceding state via the bootstrap, and $\lambda = 1$ maintains an extended history and is comparable to a Monte Carlo update. Indeed, although most previous work has assumed $\lambda = 0$, eligibility traces help to speed learning over long temporal horizons. Further, arguably the most biologically plausible case of all is $\lambda = 1$, since this case approximates Hebbian learning with eligibility traces. In particular, when $\lambda = 1$, each bootstrap term is canceled on the next trial's update due to the trace itself decaying by exactly $\gamma$. This means it can be omitted and an equivalent update for each state can be computed using just the eligibility traces and the encountered states. This amounts to a series of scalar, element-wise updates for each state encounter weighted by each state's decaying eligibility, or equivalently Hebbian learning between the decaying eligibility traces and the encountered states. In contrast, the bootstrap rule for $\lambda < 1$ requires a high-dimensional vector of error signals spanning destination states or state features, which imposes connectivity requirements that have, for some proposals, been argued to be implausible [17,22,23].

We first compared fits to trial-by-trial reaction times by SR-TD (Fig 2A) and a number of simpler alternatives. These included a fixed SR without trial-by-trial learning (based on expected transition probabilities across the entire walk), a model that learned one-step transition probabilities (i.e., the one-step state-to-state transition matrix $T$, Fig 2B), and a recency learning rule that simply learned destination state probabilities unconditional on the predecessor state. These models were fit to reaction times across three sets of random walks, either on a modular (n=173), lattice (n=185) or random (n=177) graph, with each walk comprised of 1500 trials. Reaction times were fit via a shifted log-normal distribution, $\log(rt_t - shift) \sim N(\mu_t, \sigma^2)$ where $rt_t$ is the reaction time on trial $t$, $shift$ is the minimum possible RT predicted by the model (fit as a fraction between 0 and 1 of that subject's minimum observed RT), and $\mu_t$ is a predicted value for trial $t$ given by one of the respective models, which in the case of the SR was assumed to be proportional to its future occupancy weight $M_{s_{t-1}, s_t}$ plus the weighted sum of a number of additional nuisance variables (see Methods). In order to compare the dynamics of different learning rules while excluding the possibility that these might interact with arbitrary assumptions about initialization of $M$, the first 500 trials of each walk were used to equilibrate the model and to establish model representations, but did not contribute to likelihood. Note that due to stochastic state encounters, all the learning

PLOS Computational Biology

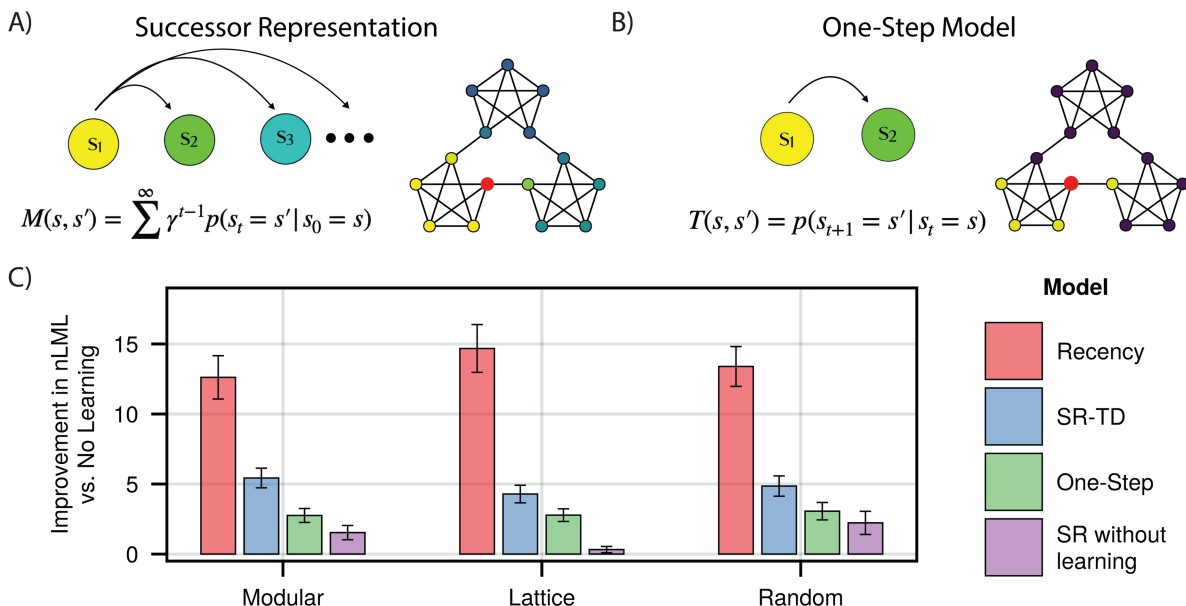

**Fig 2**. **Learning models. A) and B)** Successor Representation and One-Step Learning Rule. The SR predicts multiple steps into the future, while the one-step model predicts only the next node. Shown are respective predictive weights from the red node, when both models are converged. **C)** Model fits for each subject group. Indicated is the per-subject improvement in approximated negative log marginal likelihood versus a baseline model which includes no prediction of upcoming nodes, where a higher value indicates a better fit. Error bars indicate +/- 1.96 standard error.

models predict different patterns of fluctuations around the learned representation even in the steady state, and it is these we wish to examine to isolate learning dynamics separate from initialization.

We found that the SR-TD model provided a significantly better fit than did the one-step model (Paired $t$-tests, one-sided $t_{172} = 11.28$, 95% CI: 2.21 to 3.14 (modular), $t_{184} = 11.08$, 95% CI: 1.24 to 1.78 (lattice), $t_{176} = 10.34$, 95% CI:1.45 to 2.14 (random), $p$-values <0.001) to all sets of subjects, suggesting that, in-line with previous analyses, reaction times are reflective of the SR (Fig 2C). Moreover, the trial-by-trial learning model provided a significantly better fit than did the SR without learning: fits which incorporated trial-by-trial learning outperformed those with a static, ground-truth knowledge of the underlying graph structure (Paired $t$-tests, one-sided $t_{172} = 9.82$, 95% CI: 3.11 to 4.68 (modular), $t_{184} = 12.29$, 95% CI: 3.33 to 4.60 (lattice), $t_{176} = 6.28$, 95% CI: 1.80 to 3.45 (random), $p$-values <0.001), indicating that we are capturing trial-by-trial transients even after the equilibration period. To our surprise, however, the recency model outperformed SR-TD (Paired $t$-tests, one-sided $t_{172} = 6.28$, 95% CI: 5.91 to 8.46 (modular), $t_{184} = 14.52$, 95% CI: 8.98 to 11.81 (lattice), $t_{176} = 14.22$, 95% CI: 7.35 to 9.72 (random), $p$-values <0.001). In other words, a model learning unconditional recency-weighted node frequency explained more variance in trial-by-trial RTs than any of the higher-order models, raising a question of whether people were in fact learning the SR. This was the case even though, following earlier work, the SR models incorporated nuisance variables which were included to control for such recency (but were evidently insufficiently flexible to do so; see Methods).

## SR learning exists alongside recency learning

Next, we asked whether behavior was best explained by recency learning alone, or whether further higher-order learning effects were present alongside it. Converging evidence via other approaches suggest that people learn predictive representations in such contexts [1,2,24], and thus, a likely explanation is that recency-driven expectations (e.g., unconditional prior probabilities over states) exist alongside higher-order state-to-state conditional representations, with differing

degrees of impact on reaction times. We thus combined the models tested previously, so as to simultaneously (but independently) learn recency predictions alongside higher-order effects, with different learning rates and $\beta$ weights for the two components. We then compared the fit of these models to one another, as well as to the model of recency learning alone.

We observed that models which included higher-order predictions displayed improved fits across all subject groups, when compared to the recency model (Fig 3A). Fits of the SR-TD model consistently provided the best fit in all subject groups, with differences from the one-step model being statistically significant in all subject groups (Paired *t*-tests, one-sided $t_{172} = 3.89$, 95% CI: 0.50 to 1.54 (modular), $t_{184} = 4.49$, 95% CI: 0.18 to 0.47 (lattice), $t_{176} = 5.06$, 95% CI: 0.39 to 0.89 (random), *p*-values <0.001), and differences from the recency-alone model being statistically significant in all subject groups as well (Paired *t*-tests, one-sided $t_{172} = 7.12$, 95% CI: 1.45 to 2.56 (modular), $t_{184} = 10.34$, 95% CI: 1.51 to 2.23 (lattice), $t_{176} = 7.68$, 95% CI: 1.50 to 2.54 (random), *p*-values <0.001). Additionally, the difference from an SR model without learning was significant for lattice and random graphs (Paired *t*-tests, one-sided $t_{172} = 1.07$, 95% CI: -0.23 to 0.79, *p*<0.28 (modular), $t_{184} = 7.43$, 95% CI: 1.09 to 1.88, *p*<0.001 (lattice), $t_{176} = 2.21$, 95% CI: 0.07 to 1.12, *p*<0.03 (random)). We further verified that parameters in this model were recoverable (Fig A in S1 Text). Thus, while recency learning drives behavior more strongly than the higher order effects, it does not fully explain reaction time biases. Instead, those biases are best explained by a combination of the recency learning and a learned multi-step predictive model.

To further verify that our effects were not solely explained by recency, we examined a prediction of the SR in the modular graph which is not shared by a recency learning model, the difference between first and second trials upon entry to a new cluster. For the modular graph, the SR assigns a lower predictive weight to cross-cluster transitions than to transitions within the same cluster. However, once a new cluster is entered, the SR now will exhibit a bias towards predicting

A)

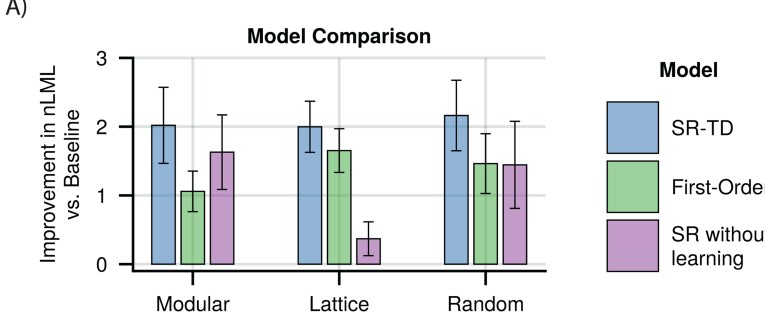

B)

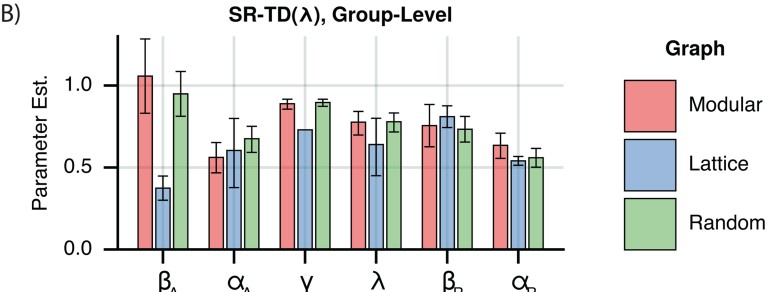

**Fig 3. Combined learning models. A)** Model fits for combined models for each graph. Indicated is the per-subject improvement in approximated negative log marginal likelihood versus the recency model, where a higher value indicates a better fit. Error bars indicate +/- 1.96 standard error. **B)** Group-level parameter estimates for the best fitting model, which combined recency and SR-TD($\lambda$) learning. Error bars indicate 95% CI. $\beta_A$ and $\alpha_A$ indicate $\beta$ weight and learning rate for the SR-TD($\lambda$) component. $\beta_W$ and $\alpha_W$ indicate $\beta$ weight and learning rate for the recency component.

nodes in the new cluster. On average, this suggests that the first node should exhibit a slower RT than the second node. We fit a mixed effects linear model to residual log RTs after accounting for motor response and trial number, and included recency as a categorical control (see Methods). We found that the second trial was indeed significantly faster than the first ($\beta$ = -0.029, SE = 0.013, t(102.0) = -2.33, p = 0.02), confirming that learners exhibit behavior predicted by the SR.

We next examined the fit parameters to understand the contribution of different learning mechanisms to our results (recovered subject-level estimates are shown in Figs B–G in S1 Text, and pairwise scatter plots are shown in Figs H–M in S1 Text). In addition to a $\beta$ weight that scales the contribution of the SR expectancies onto reaction times, SR-TD is parameterized by $\gamma$, which controls how far into the future predictions extend, and $\lambda$ which controls the duration of eligibility traces. Two special cases are of particular interest. First, $\lambda = 0$ (bootstrapping alone) is the case most often assumed previously. At the other extreme, $\lambda = 1$ relies heavily on long-lasting eligibility traces (and indeed admits of Monte Carlo variants that can be implemented with only Hebbian learning and no bootstrapping at all). We examined estimates from our combined model, and found that $\lambda$ was consistently estimated quite high (modular: 0.752, 95% CI: [0.682, 0.812], lattice: 0.637, 95% CI: [0.448, 0.798], random: 0.786, 95% CI: [0.725, 0.838]) but that the extreme values of 0 and 1 were excluded (Fig 3B). This suggests, in contrast to previous work, that learning is predominantly driven by eligibility traces. To test whether eligibility traces alone explained learning, we first verified that a model with freely varying $\lambda$ should be distinguishable from one with $\lambda$ fixed at extreme values of 0 or 1 (Fig N in S1 Text). We then compared subject-level marginal likelihoods (see Model Comparison) from our model fits with $\lambda$ as a free parameter to those where $\lambda$ was fixed at 1, and found that allowing it to freely vary significantly improved model fits in all cases (Paired $t$-tests, one-sided $t_{172} = 3.32$, 95% CI: 0.14 to 0.55, $p<0.002$ (modular), $t_{184} = 5.57$, 95% CI: 0.15 to 0.31, $p<0.001$ (lattice), $t_{176} = 3.68$, 95% CI: 0.14 to 0.46, $p<0.001$ (random)). This suggests, in contrast to previous work, that learning is predominantly driven by eligibility traces, though we do not rule out bootstrapping altogether.

## Human RTs exhibit signatures of trace updating but not bootstrapping

We next sought to more directly detect model-agnostic signatures in the data of eligibility trace and bootstrap learning mechanisms. The key signature of trace learning is that when a state sequence such as ABC is visited, the prediction from A is immediately updated to expect C. (Conversely, bootstrapping without eligibility traces requires a second AB transition to chain expectation from C back to B then B back to A.) This formulation suggests that we could look for evidence of reaction times being modulated in this way, by finding sequences of trials that differ only by whether a trace update could have occurred, and testing whether reaction times systematically differ between those sequences.

In particular, we compared reaction times for the final elements of sequences of the form '$SXTST$' versus '$X_1X_2TST$', which measures expectancy of a target T on its second visit, following the second visit to a source S (in the first case) or following the first visit to S (in the second case). If a trace is used for learning, then expectancy of T following S is increased by the initial sequence '$SXT$', which is tested by the comparison of '$SXT$' vs '$X_1X_2T$' (Fig 4A). We directly tested for this effect by finding all sequences matching the above patterns, and fitting a hierarchical mixed-effects model to subject reaction times (see Methods), alongside simulated eligibility trace and bootstrap learner agents. As predicted, we observed that participants were significantly faster at responding to the final element of sequences of the first type (mixed-effects model, $\mu = 30$ms, SE=0.01, $z = 2.93$, $p<0.0034$), suggesting that they updated their expectations in a trace-like manner (Fig 4B).

Next, we tested a similar signature for a bootstrap backup, the central feature of TD(0). For this purpose, we contrasted reaction times from sequences matching '$BTSBST$' to those matching '$XTSBST$'. In the first sequence, the transition from a bootstrap node B to the target T at the start increases the prediction of T given B. The subsequent transition from S to B should, in turn, chain that prediction of T from B to S (an effect not occurring in the second sequence as the transition from B to T was never initially increased), and thus decrease reaction times for the final transition from S to T (Fig 4C). We observe such a difference in reaction times for simulated bootstrap agents, but not simulated eligibility trace

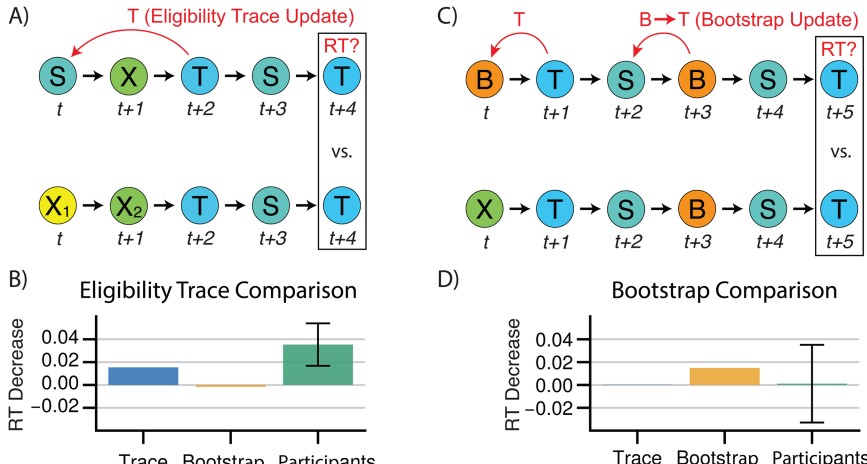

**Fig 4**. **Model agnostic signatures of trace updating and bootstrapping. A)** The sequence $S \rightarrow X \rightarrow T \rightarrow S$ increases the weight given to $p(T|S)$ for trace learners but not bootstrap learners. The sequence $X_1 \rightarrow X_2 \rightarrow T \rightarrow S$ is provided as a recency-matched control. **B)** Trace effect for simulated trace and bootstrap learners learners. Participants exhibit a significant trace effect. Error bars indicate +/- 1.96 standard error. **C)** The sequence $B \rightarrow T \rightarrow S \rightarrow B \rightarrow S$ increases the weight given to $p(T|S)$ for bootstrap learners but not trace learners. The sequence $X \rightarrow T \rightarrow S \rightarrow B \rightarrow S$ is provided as a recency-matched control. **D)** Bootstrap effect for simulated trace and bootstrap learners learners. Participants do not exhibit a significant trace effect. Error bars indicate +/- 1.96 standard error.

agents. However, we detect no significant effect of sequence type for reaction times among participants, in contrast to our eligibility trace effect (mixed-effects model, $\mu$=3.5ms, SE=0.219, p<0.87, Fig 4D). While this may be consistent with the high estimated $\lambda$ from the model-based analysis, negative results should be interpreted with caution. Here, compared to the eligibility trace effect, the increased pattern length (six elements instead of five) means that we have fewer observations — 15,493 across all subjects, versus 94,000, limiting our power compared to the eligibility trace analysis. The standard error on the estimate (reflecting this reduced power) implies that this analysis cannot confidently rule out an effect on RTs roughly the size of that estimated for the eligibility trace.

## Discussion

It has long been known that reaction times in sequential response tasks like graph learning are facilitated by learned expectations [25–30]. More surprisingly, it has recently been shown that these predictions span multiple future timesteps [11–13,31–33]. In this respect, they coincide suggestively with a form of temporally abstracted prediction model called the successor representation, which has also been a focus of recent study in other areas of decision making [4–6,10,34], cognitive map learning [24,35–39], and memory [33,40]. These RT effects thus might offer a novel window into the mechanisms by which such long-run predictions are formed over delays, which is one of the key computational challenges in making effective decisions but challenging to study in many of the other settings where the SR arises. Here we investigate the trial-by-trial learning rules driving predictive effects in graph learning.

Across several different graph structures and using both model fits and model-agnostic signatures, we find evidence for a predominant role of eligibility traces in such learning. While not the rule most often assumed, these results are in line with prior suggestions that traces are a biologically plausible mechanism to enable the learning of predictive associations over delays [18,20,22].

### Reaction times are strongly driven by recency learning effects alongside SR

One surprising aspect of the current results was that, although SR learning is significantly demonstrable, it exists alongside more basic learning effects. Specifically, RTs were strongly driven by recency learning, i.e., a vector of state

probabilities derived from a running average over recent experience but unconditional on the predecessor state. In fact, considered in isolation, these recency learning effects provide a better prediction of RTs than do higher-level models alone. This was true even though our SR models contained nuisance variables intended to control for recency effects, which were evidently insufficiently flexible to capture the timescale of recency. Thus, properly accounting for these effects is crucial, particularly if we hope to characterize higher-order learning.

Moreover, this result suggests a more comprehensive model of learning than previously proposed. Of course, participants have no reason to assume that the states they encounter are drawn from a graph (i.e., Markov according to one-step conditional probabilities), but instead likely entertain multiple predictive models. In this respect, the recency model is the simplest case of the idea that people and animals learn predictive models across multiple timescales [41,42], in this case no timescale at all. This ability to learn models across multiple timescales may be supported, in part, by range of timescales of dopaminergic signaling observed in animal studies [43–45].

While statistical patterns akin to this recency effect are ubiquitous in the natural world (generally speaking, the recent past is an excellent predictor of the future), people are capable of learning other patterns such as an "anti-recency effect" [46] after sufficient exposure. Why, then, do we specifically see such strong recency behavior on the current task? One possibility is that people rely on priors which suggest the form of recency we observe, and unlike [46], are not given evidence to the contrary. Another is that they are quickly adapting to the observed task statistics. Such adaptation would likely be at an extremely short timescale. Indeed, we do observe a much larger improvement in reaction times at the start of the task, but this is likely confounded by learning the task response paradigm itself. In the future, we hope to test to what degree this recency model is drawn from prior experience versus learned directly from the statistics of the task at hand.

## Mechanisms for learning long-run predictions

The SR predicts aggregated, discounted future long-run encounters of each state (here, graph node) conditional on the current state. Such predictions are of particular interest in reinforcement learning, because they may be used to forecast the long-run consequences of candidate actions and guide choices toward later rewarding states. In the present task, where movements are instructed rather than chosen, such predictions may more implicitly drive motor preparation, thereby facilitating faster RTs. However, in this setting it is less clear why long-run—rather than next-step conditional—expectancies should be used. This may reflect the brain learning predictions optimized to other, more general settings (e.g., non-Markovian dynamics or sequential choice tasks) rather than graph learning *per se*, or it may reflect the learning of a one-step model that is, in effect, regularized by longer-run expectancy [13]. Another possibility comes from motor sequence learning studies, where people are shown not only the next response cue but a larger window of upcoming cues. Participants can leverage such future information to speed their responses[47], likely by preparing actions multiple steps into the future[48]. Such preparatory behavior in the current task could be supported by learning long-run expectancies with the SR.

Regardless, similar to the more studied case of temporal credit assignment (i.e., predicting long-run future rewards; [19, 49–53]), learning such predictions requires spanning temporal gaps to learn associations between graph nodes encountered at a delay. Also as in the case of rewards, three distinct, though not mutually exclusive, classes of mechanisms have generally been proposed for learning an SR [15]: chaining predictions successively by bootstrapping, spanning temporal gaps by maintaining sustained eligibility traces, and repeatedly iterating a one-step model to increase the timescale of prediction.

In the present study, we focus on the first two of these (see below for more on iteration), which can be hybridized by a single parameterized learning rule, SR-TD($\lambda$). This form is particularly convenient for descriptive estimation because it casts the mechanism question as that of estimating the parameter $\lambda$, enabling us to identify mixtures of both but also detect nested special cases $\lambda = 0$ corresponding to pure bootstrapping and $\lambda = 1$ similar to pure eligibility traces.

Most work on the SR has tended to envision that its predictions are learned by bootstrapping, i.e. SR-TD(0) [14,33,38]. Here, each time a state is encountered, its own forward-looking predictions are "backed up" to its immediate predecessor state, with a net effect that thus accumulates and chains backward over multiple traversals. This is accomplished using temporal-difference prediction errors, and the popularity of the algorithm is probably in part due to evidence that the brain's dopamine system appears to signal such bootstrap errors for the case of chaining reward expectancy [54–56]. However, whereas reward is a scalar and has been argued to be appropriate for a relatively homogeneous dopamine projection, it is less clear whether dopamine (or if not, what other system) can deliver the higher-dimensional set of vector-valued prediction errors required to bootstrap individual SR predictions for many successor states [17,22].

A second mechanism for learning associations at a delay is to maintain "eligibility traces" for each state that increment when it is encountered and gradually decay over time [57]. These can operate in isolation, standing in for presynaptic activity in a Hebbian learning rule to associate previously encountered states with each newly encountered one. They can also work in concert with bootstrapping; in particular, for $\lambda > 0$, SR-TD($\lambda$) uses such traces to transport each bootstrap-based update over multiple preceding timesteps. In the extreme case of SR-TD(1), learning is essentially driven only by traces and similar to the simpler Hebbian rule, and the bootstrap (though still algebraically present) contributes only transiently and incidentally. Traces have been argued to be neurally plausible (for the SR, arguably moreso than bootstrap), in that eligibility might be maintained by some persistent chemical change locally at a synapse or spine [58–60]. Traces could also be subserved by persistent patterns of activity, as exponentially decaying neuronal firing is observed in the cortical neurons [61]. Such firing is thought to express a continuous range of time constants, which could provide a basis for learning SRs across multiple temporal scales [42]. Alternatively, neurons representing eligible states might be reactivated after a delay, via history stored in some separate memory system [18]. The maximum entropy graph learner envisioned by [13] also works by reactivating recently active predecessor states, and corresponds (at least in expectation) to a trace update.

Accordingly, in our model fitting analyses, we focused on estimating $\lambda$ (alongside other parameters and nuisance effects) to detect evidence for bootstrapping and/or traces. Best-fit values tended to be intermediate, though closer to 1 than 0. Taken at face value (since both extreme models are rejected), these results support a contribution of both mechanisms. In conjunction with evidence for traces also in our separate model-agnostic analyses (comparing RTs between different state sequences chosen to expose the effects of either bootstrap or traces), we take our results as clear evidence of a strong role for traces.

We are more cautious about the evidence in favor of bootstrapping, due to high (albeit sub-unity) estimated $\lambda$ and also the failure of our model-agnostic analyses to detect positive evidence for it. Interestingly, this negative result stands in contrast to our recent clearer detection of bootstrapping using somewhat similar analyses (albeit of neural data) in a reward learning task, where TD(0) is also a more neurally plausible algorithm [19]. Of course, this difference may reflect lower power in the model-agnostic analyses for bootstrap relative to traces (due to fewer, longer sequences required to expose the former). Another possibility is that a better fitting trace-only learning rule remains to be discovered: SR-TD(1) can be learned through a variety of methods that rely on strictly Hebbian trace rules instead of bootstrapping, due the feature that subsequent bootstrap terms will cancel each other out. These variations in turn all exhibit distinct transients, and further work is required to investigate whether any of these might plausibly provide a better explanation of behavior than SR-TD($\lambda$).

Such a Hebbian learning rule would also relate to another prominent model of human memory, the TCM (Temporal Context Model) [62]. The TCM was developed to explain a number of phenomena consistently observed in free recall, including recency and contiguity effects. Interestingly, the TCM can be understood as learning the SR [40,63] so long as stimuli are fully unique, but the two diverge upon any repetition (such as re-visiting the same node in the current study). Specifically, under certain constraints, the context vector in the TCM functions identically to the eligibility trace in the SR, where the first visit to state $j$ will set the prediction from $i$ to $j$ to $e_i$. Due in part to nonzero contribution of bootstrapping,

the TD($\lambda$) rule for learning an SR differs from classic TCM on the second encounter [63], whereas a purely Hebbian (eligibility trace only) variant of TD(1) SR learning stays closer to the original conception of TCM. Unlike classic list learning experiments (which typically present each item only once), the present experiments visit each graph node many times, exercising the differences between these various learning approaches, which largely coincide on the first visit. Of course, it remains to be tested whether similar conclusions (e.g. that learning is best explained by a combination of traces and bootstrap, but primarily the former) would be seen in analogous experiments testing memory recall, in experiments with multiple item encounters.

Finally, we also considered, but do not report here, "SR-MB" learning rules [15], which estimate a simple one-step conditional state-state probability model $T$ via Hebbian updates, i.e., with neither traces nor bootstrap. These can be iteratively accumulated after each update (equivalent to a sum of matrix powers [64], or a matrix inversion $M = (I - \gamma T)^{-1}$) to recover the long-run SR [65]. This represents the third general route to long-run SR-prediction, analogous to iterative model-based rollouts in the reward-learning case. We omit these models from the analyses reported here, because they did not capture any unique variance in RTs, suggesting no detectable contribution of SR-MB. Instead, in all cases, we estimated a nested special case ($\gamma \to 1$) that is, for this model, equivalent to the recency learner.

### Temporal discounting and choice

Alongside the parameter $\lambda$, which governs how long eligibility persists for learning over temporal delays, we also estimate a second parameter, $\gamma$, which controls how far into the future state occupancy predictions accumulate. In choice tasks, $\gamma$ has long been of interest because it controls intertemporal preferences, i.e., how reward amounts trade off against delays. In the current task (which involves no choices), the parameter instead governs the timescale of implicit predictions as reflected in RT facilitation. Interestingly on the average over participants, we estimate all values falling between $\gamma = .717$ and $\gamma = 0.894$, equivalent to a temporal horizon of 3.5 to 9.4 timesteps. An interesting question for future work is whether the predictive model we measure in RTs is also used for choice, e.g. if subjects learn a graph and are subsequently asked to plan routes or choose between states for reward [34,66]. In this latent learning case, $\gamma$ estimated from RTs might reflect individual differences that also play out in intertemporal choice. Alternatively, $\gamma$ might be tuned to particular environments, e.g. the degree to which certain topologies enable longer-range prediction.

## Methods

### Ethics statement

All participants provided written informed consent as specified by the Institutional Review Board of the University of Pennsylvania, and study methods and experimental protocols were approved by the Institutional Review Board.

### Behavior

We fit our models to participants from [12]. Participants completed 1500 trials of a serial reaction time-like task, responding to a cue shown on the screen as rapidly as possible with a one- or two-button key combination. Unbeknownst to participants, the sequence of cues followed a traversal through one of various graph structures: either a modular graph composed of 15 nodes in 3 clusters, a lattice graph, or a randomly connected graph. Each subject was assigned two of the three graphs, and performed the task twice in sequence with counterbalanced ordering. Node sequences were generated via a random walk on the assigned graph. An additional subset of subjects performed the task on one of the three aforementioned graphs, followed by 500 trials on a fully connected graph. The assignment of key combination to node was randomized across subjects. We include the first set of trials for those subjects, but exclude the fully connected graph from our analysis. We excluded all trials on which subjects responded incorrectly, as well as trials with implausible RTs (those less than 30 ms). Across the two task stages of 1500 trials each, our analysis included 173 subjects for the modular graph, 185 subjects for the lattice graph, and 177 subjects for the set of random graphs.

## Modeling approach

To compare proposed learning algorithms that would give rise to observed biases in reaction times, we estimated the likelihood of the sequence of RTs given the respective learning algorithm under a shifted log-normal distribution, which takes the form

$$\log(rt_t - shift) \sim N(\mu_t, \sigma^2), \tag{1}$$

where $rt_t$ is the reaction time on trial $t$, *shift* is the minimum possible RT predicted by the model (fit as a fraction between 0 and 1 of that subject's minimum observed RT), and $\mu_t$ is a predicted value for trial $t$ given by one of the respective models. Trial value was rescaled to vary between 0 and 1. Per-subject parameters were fit hierarchically over the group via expectation maximization, and models were compared via approximated negative log marginal likelihood. For simplicity, group-level covariance was modeled as the identity matrix. To focus solely on transient trial-by-trial variation in prediction, and avoid artifacts reflecting the initialization of the models, likelihood was maximized only for the last 1000 trials in each sequence—the first 500 trials were simulated, but the likelihoods of the associated reaction times were not included for fitting purposes. Models were fit to data from task stages for a given graph, with additional group-level parameters fit to the difference between the stages via a covariate set to -1 for the first stage, and 1 for the second stage.

## Recency learning model

In our recency learning model, reaction times were a function of individual node probabilities, where $\mu_t$ was defined as

$$\mu_t = \mu_0 + \beta_{trial}t + target_{2-15} + \beta_A W[s_t] + \beta_{ntrials}r_{ntrials} + \beta_{lag10}r_{lag10}, \tag{2}$$

where $\mu_0$ is a baseline RT, *trial* is trial number to account for non-specific learning effects, $target_{2-15}$ are fits for the various finger combinations, and $W[s_t]$ was a function of the upcoming node. Each entry of $W$ was initialized to 1/15, and W was updated on each trial as follows:

$$W \leftarrow (1 - \alpha)W + \alpha I(s_t), \tag{3}$$

where $I(s_t)$ is a one-hot vector at position $s_t$.

Additionally, a simpler recency model was added in-line with previous work by including regressors $\beta_{ntrials}$ and $\beta_{lag10}$, which respectively were applied to the number of trials elapsed since node $s_t$ last occurred, and the log of the number of occurrences of node $s_t$ within the last 10 trials.

## One-step learning model

In our one-step transition model, reaction times were a function of pairwise transition probabilities, where $\mu_t$ was defined as

$$\mu_t = \mu_0 + \beta_{trial}t + target_{2-15} + \beta_A T[s_{t-1}, s_t] + \beta_{ntrials}r_{ntrials} + \beta_{lag10}r_{lag10}, \tag{4}$$

where $\mu_0$ is a baseline RT, *trial* is trial number to account for non-specific learning effects, $target_{2-15}$ are fits for the various finger combinations, and $T[s_{t-1}, s_t]$ was the entry of the learned transition matrix from the previous to upcoming node. Each entry of T was initialized to 1/15, and T was updated on each trial as follows:

$$T[s_{t-1}, :] \leftarrow (1 - \alpha)T[s_{t-1}, :], \tag{5}$$

$$T[s_{t-1}, s_t] \leftarrow T[s_{t-1}, s_t] + \alpha. \tag{6}$$

## SR-TD learning model

Our SR-TD model learned the SR via a TD($\lambda$) update rule, and reaction times were a function of the predicted future occupancy of node $s_{t+1}$ given node $s_t$. $\mu_t$ was defined as

$$\mu_t = \mu_0 + \beta_{trial}t + target_{2-15} + \beta_A M[s_{t-1}, s_t] + \beta_{ntrials}r_{ntrials} + \beta_{lag10}r_{lag10}, \qquad (7)$$

where $\mu_0$ is a baseline RT, *trial* is trial number to account for non-specific learning effects, $target_{2-15}$ are fits for the various finger combinations, and $M[s_{t-1}, s_t]$ was the SR's learned $M$ matrix, normalized by the sum of $M[s_{t-1}, :]$. $e$ was implemented as a dutch trace [67]. M was initialized to $(I - \gamma T)^{-1} * T$ for a uniform transition matrix $T$, and updated on each trial as follows:

$$
\begin{aligned}
&e[s_{t-1}] \leftarrow (1 - \alpha)e[s_{t-1}] + 1 \\
&\delta = I[s_t] + \gamma M[s_t, :] - M[s_{t-1}, :] \\
&\text{for } s' \text{ in } 1{:}15 \\
&\quad M[s', :] \leftarrow M[s', :] + \alpha e[s']\delta \\
&\text{done} \\
&e \leftarrow \gamma\lambda e,
\end{aligned}
\qquad (8)
$$

where discount factor $\gamma$ and trace parameter $\lambda$ were fit to each subject. Note that the SR as calculated here reflects a one-step-ahead variation, equivalent to $T + \gamma T^2 + \gamma^2 T^3 + \cdots$.

## SR model without learning

Our static SR model calculated the SR for the empirical transition probabilities across the entire sequence for a given subject. That is, we calculated $T_{i,j} = \frac{\sum_t (s_{t-1}=i)(s_t=j)}{\sum_i (s_{t-1}=i)}$, and then $M = T * (I - \gamma T)^{-1}$. $\mu_t$ was defined as

$$\mu_t = \mu_0 + \beta_{trial}t + target_{2-15} + \beta_A M[s_{t-1}, s_t] + \beta_{ntrials}r_{ntrials} + \beta_{lag10}r_{lag10}, \qquad (9)$$

where $\mu_0$ is a baseline RT, *trial* is trial number to account for non-specific learning effects, $target_{2-15}$ are fits for the various finger combinations, and $M[s_{t-1}, s_t]$ was an entry of the static SR's $M$ matrix.

## Combined models

For models that were a combination of recency learning and a higher-order learning model, $\mu_t$ was simply given as

$$\mu_t = \mu_0 + \beta_{trial}t + target_{2-15} + \beta_W W[s_{t-1}, s_t] + \beta_A A(s_{t-1}, s_t) + \beta_{ntrials}r_{ntrials} + \beta_{lag10}r_{lag10}, \qquad (10)$$

where $A(s_{t-1} + s_t)$ was $M$ for the SR or $T$ for the one-step model.

## Parameter estimation

We optimized the free parameters of the learning algorithms by embedding each of them within a hierarchical model to allow parameters to vary between subjects or simulated sessions. Subject-level parameters were modeled as arising from a population-level Gaussian distribution over subjects. We estimated the model, to obtain best fitting subject- and group-level parameters to minimize the negative log likelihood of the data using an expectation-maximization algorithm with a Laplace approximation to the session-level marginal likelihoods in the M-step [68].

## Model comparison

To compare models which may have different numbers of free parameters, correcting for any bias due to overfitting, we marginalized the participant-level parameters by computing a Laplace approximation to the negative log marginal likelihood of each participant's data. Because the participant-level parameters were themselves estimated in the context of a hierarchical model governed by additional population-level parameters, we added an additional AIC penalty for these parameters, divided equally between subjects. This provides a fit score for each subject and model. Finally, we used paired t-tests on these scores across participants, between models, to formally test whether a model fit significantly better than another over the population [69].

## Novel cluster analysis

To test whether participants exhibited slower responses on the first node upon entering a new cluster than the second, we fit a linear mixed effects model to residualized RTs. These residualized RTs were derived by fitting a model of the form of our learning models, but which only included effects of trial number and motor response. We then found all trials that were either the first or second entry into a new cluster. We fit a model of the form *resid_rt ~ time_since_node + cluster_step + (1 + cluster_step|subject))*, where *cluster_step* was either 0 for the first node or one for the second, and *time_since_node* was the number of trials since the node was last seen, for all lags up to 100. *time_since_node* fit as a categorical with an effects coding scheme such that *cluster_step* was relative to the grand mean. Models were fit *MixedModels* v4.24.1 [70] in *Julia* 1.10.4 [71].

## Trace analysis

We sought to test in a model-agnostic fashion whether peoples' expectations reflected a trace update—that is, whether a transition expectation $S \rightarrow T$ was strengthened even when a third node X occurred between S and T as in $S \rightarrow X \rightarrow T$. To do so, we looked for sequences of the form 'SXTST' for any three distinct nodes $S$, $X$, and $T$, and took the reaction time on the final $T$ as reflecting $S \rightarrow T$ after a trace update. We tested these reaction times against those from the final $T$ of sequences of the form '$X_1 X_2 TST$', in which recency of $T$ is controlled for, but a trace update would not have occurred for $S$.

We then fit a hierarchical linear mixed model using *MixedModels* v4.24.1 [70] in *Julia* 1.10.4 [71] to these reaction times, where *rt* was predicted via

$$rt \sim \mu + \text{is\_trace} + \text{target}_{2-15} + (1 + \text{is\_trace} \mid \text{graphset}) + (1 + \text{is\_trace} \mid \text{subject}), \qquad (11)$$

where is_trace indicates RTs were from 'ABCAC' sequences (versus 'ABCDC'), *target*$_{2-15}$ are fits for the various finger combinations, and random effects were included for intercept and is_trace for each subject and graph instance (Modular/Lattice/Random, first or second walk). The model was fit across all subjects.

## Bootstrap analysis

Our model-agnostic test for a bootstrap update was performed similarly, but for a sequence designed to contrast bootstrap vs. non-bootstrap effects on reaction times. 'BTSBST' and 'XTSBST'. In 'BTSBST', the initial sequence 'BT' strengthens the prediction of T from B. For a bootstrapped update, 'SB' will then cause S's predictions to shift towards those of B, meaning 'ST' will be strengthened as well, leading to an enhanced RT on the final step of 'ST'. In the second sequence, the initial 'BT' is replaced by 'XT', and thus 'ST' should not be systematically upwardly biased by a bootstrap update.

We then fit a hierarchical linear mixed model using *MixedModels* v4.24.1 [70] in *Julia* 1.10.4 [71] to these reaction times, where *rt* was predicted via

$$rt \sim \mu + \text{is\_bootstrap} + \text{target}_{2-15} + (1 + \text{is\_bootstrap} \mid \text{graphset}) + (1 + \text{is\_bootstrap} \mid \text{subject}), \qquad (12)$$

where is_bootstrap indicates RTs were from 'ABCAC' sequences (versus 'ABCDC'), $target_{2-15}$ are fits for the various finger combinations, and random effects were included for intercept and is_bootstrap for each subject and graph instance (Modular/Lattice/Random, first or second walk). The model was fit across all subjects.

## Supporting information

**S1 Text. Contains supplemental text, methods and figures.**
(PDF)

## Author contributions

**Conceptualization:** Ari E. Kahn, Nathaniel D. Daw.

**Data curation:** Ari E. Kahn.

**Formal analysis:** Ari E. Kahn.

**Funding acquisition:** Nathaniel D. Daw.

**Investigation:** Ari E. Kahn.

**Methodology:** Ari E. Kahn, Dani S. Bassett, Nathaniel D. Daw.

**Project administration:** Ari E. Kahn.

**Resources:** Dani S. Bassett, Nathaniel D. Daw.

**Supervision:** Dani S. Bassett, Nathaniel D. Daw.

**Visualization:** Ari E. Kahn.

**Writing – original draft:** Ari E. Kahn.

**Writing – review & editing:** Ari E. Kahn, Dani S. Bassett, Nathaniel D. Daw.

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
