## [Decision Letter · Decision Letter 0]

13 May 2025

PCOMPBIOL-D-25-00267

Trial-by-trial learning of successor representations in human behavior

PLOS Computational Biology

Dear Dr. Kahn,

Thank you for submitting your manuscript to PLOS Computational Biology. After careful consideration, we feel that it has merit but does not fully meet PLOS Computational Biology's publication criteria as it currently stands. Therefore, we invite you to submit a revised version of the manuscript that addresses the points raised during the review process.

Please submit your revised manuscript within 60 days Jul 13 2025 11:59PM. If you will need more time than this to complete your revisions, please reply to this message or contact the journal office at ploscompbiol@plos.org. Please include the following items when submitting your revised manuscript:

We look forward to receiving your revised manuscript.

Kind regards,

Wolfgang Einhäuser

Academic Editor

PLOS Computational Biology

Lyle Graham

Section Editor

PLOS Computational Biology

**Additional Editor Comments :**

While the reviews are positive in general, they also raise some major concerns, which need to be addressed. Some of the presentation of the Methods is overly concise, an appendix or supplement might help for issues the authors do not want to include in the main text, but which would be needed for replication. I in particular had trouble understanding the human experiments from the paper. I understand that the data is from a previous publication, but at least the group sizes (subjects per group) and the number of trials per subject would be useful and also help understanding whether the statistical tests conducted were appropriate. When I tried to access the OSF repository, I was required to send a personalized access request. While this would have been ok for me, it might compromise anonymity of the reviewers, so it cannot be expected that they consider this material. The authors should consider providing a hidden osf read-only link for review of a revision (and of course make the link public, once the manuscript is accepted) or double check that their repository is public.

**Journal Requirements:**

4) Please ensure that all Figure files have corresponding citations and legends within the manuscript. Currently, Figure 1 in your submission file inventory does not have an in-text citation. Please include the in-text citation of the figure.

5) Thank you for stating "Data and code that support this study are available at https://osf.io/jupbz/." Please note that, though access restrictions are acceptable now, your entire minimal dataset will need to be made freely accessible if your manuscript is accepted for publication. This policy applies to all data except where public deposition would breach compliance with the protocol approved by your research ethics board. If you are unable to adhere to our open data policy, please kindly revise your statement to explain your reasoning and we will seek the editor's input on an exemption. 

**Reviewers' comments:**

Reviewer's Responses to Questions

Reviewer #1: In this manuscript, Kahn et al. present a series of model-based and model-agnostic analyses of previously collected data, in order to test how participants update expectations of upcoming stimulus-action requirements ("states"). They consider three main mechanisms through which participants may learn these expectations, being a "recency" mechanism, a "one-step" prediction mechanism, and a "successor representation" which is able to learn multi-step predictions of upcoming states.

The authors define a previously unconsidered model of recency and demonstrate that this is a major driver of human response times, making an important methodological contribution to the modeling of response times in graph learning tasks. They then demonstrate how multistep predictions can still further improve the model, suggesting successor representation-like learning does take place in these tasks.

Finally, the authors define two model-agnostic tests, to directly test the influence of eligibility traces and bootstrap updating, two learning mechanisms that may contribute to successor representation learning. They only find evidence for eligibility traces, but not for bootstrapping, questioning whether successor representation learning is based on vector prediction error signals as often considered in theoretical work. However, the authors note the limited amount of data available for the bootstrapping analysis compared to the trace analysis, and do not perform a test that can confirm the absence of an effect (e.g. a Bayes Factor analysis).

This work addresses an important research question, and I am particularly fond of the two model-agnostic signatures the authors were able to identify to test traces and bootstrapping. While I think the research question and results are of importance and fit the scope of Plos Computational Biology, there are several minor points that I think can be addressed to improve the strength of the current work, which I will list in order of importance:

1. Perhaps any appendices to this paper were not communicated during this peer review process, but I was missing some proof of concept (a simulation study) that the parameters and models considered in this manuscript are indeed recoverable and distinguishable (cf. Wilson & Collins, 2019; https://doi.org/10.7554/eLife.49547).

2. a. Throughout the manuscript, there are several references to the idea that an SR-TD(lambda = 1) model corresponds to a Hebbian learning rule rather than a vector-prediction-error learning rule. However, this is not mathematically exposed in the main text, and is also not readily apparent to me based on the brief model description provided in the "SR-TD learning model" section of the methods. I think readers may find it very informative if this relationship is made more explicit, and it may inform some of the points that follow.

2. b. Further detailing this, I was wondering if the SR-TD(lambda = 1) model might mimic the "temporal context model" of the episodic memory system (Howard & Kahana, 2002; https://doi.org/10.1006/jmps.2001.1388), in which a trace of previous stimuli is maintained and associated with incoming stimuli. If this is true, the authors might cite this work to better contextualize this special case.

2. c. If SR-TD(lambda = 1) is indeed such a special case, I think it is worth it to consider a variant of SR-TD in which lambda is not a free parameter, but instead fixed to 1, and to see if there is a significant improvement when lambda is allowed to be free. This would at least expose the data as containing variance that goes beyond the simple associative mechanism of the temporal context model. Currently, the authors interpret the fact that their parameter estimates do not yield this extreme value of lambda = 1 as circumstantial evidence of this, but I don't believe this is an effective way to interpret the model fit. Instead, I think it is worth statistically formalizing this point with an explicit model comparison. A lack of significance then would further question the involvement of the vector prediction error for learning multistep predictions.

2. d. In the case of SR-TD(lambda = 1), does the discount factor gamma become redundant (because learning is Hebbian and not error-driven)?

3. Could the authors point out a qualitative difference between a pure recency learning model, and a model that includes multi-step predictions? This seems to be a major point of the manuscript, and such a difference might be identifiable in the modular graph for example by considering transitions within communities as in Figure 5 of Wientjes & Holroyd (2024; https://doi.org/10.1371/journal.pcbi.1011312).

4. In the "combined models" section, the formula for mean response times no longer seems to include the ntrials and lag10 regressors. I missed a motivation for dropping these regressors from the model and wonder if some of the associated variance might not be wrongly assigned to the successor representation regressor now, potentially influencing the estimates of lambda and gamma.

5. As a purely optional and supplementary suggestion, the authors might consider whether their estimation of successor representation-related parameters (lambda, gamma) show any reliability for these participants, for example by correlating estimates of an odd-even split of the trials (contribution to the likelihood) of a single graph learning task, or even considering the parameter values of two different graph learning tasks (for the participants that completed two different graphs). Especially, if estimates of the discount factor gamma show some reliability, this could maybe be taken as evidence that prediction error-driven learning (bootstrapping?) does contribute some variance to these data.

6. Some minor typographical errors:

In the methods section "recency learning model", the beta(lag10) is not stylized into a greek letter.

In the method section describing the "bootstrap analysis", it is described to identify an enhanced RT on the final step of "SR". I believe this should refer to "ST", being the last transition of the bootstrap sequence, rather than the abbreviation for the Successor Representation (SR; causing me some confusion initially).

Reviewer #2: In this study, the authors examine response times in a serial reaction time task in which the stimuli trace out edges of a graph with one of three structures. Of particular interest, the modular graph has a temporal community structure as introduced in work by Anna Schapiro and colleagues. In previous work (e.g., Lynn, Papadopoulos, Kahn, & Bassett, 2020, Nat Physics) it was shown that human RTs are slower across community boundaries compared to within community. This finding is counterintuitive from a certain perspective; although in general RT is predicted by the entropy of the transition, the entropy of transitions within boundary are the same as across boundaries. Note that across-community edges are differentiated by powers of the transition matrix, this paper pursues the hypothesis that the successor representation (SR), which asymptotes as exponentially-weighted powers of the transition matrix, predicts human RTs. The idea is intuitively appealing; rather than ``moving'' along the true transition matrix, human learners ``move'' along the SR. The authors compare the SR to other simple models. By comparing specific sequences they can also evaluate whether the SR is updated by temporal difference learning or by simple aggregation of eligibility trace memories.

There are some substantial methodological points (below) that speak to some of the empirical basis for the conclusions about the models, but these are relatively minor. The major result, which is very likely to be sound, is that a recency-biased model carries more variance in RT than the SR model. The observation of a recency effect here does not really get much attention (recency isn't part of standard SR) but it's pretty interesting. There are certainly extremely robust recency effects in (all kinds of) memory experiments with random lists of stimuli. Recency effects among responses are also quite widespread in RT tasks (sequential dependencies in simple choice RT tasks have been well-characterized since at least the mid-1970s). However, recency bias is not universally observed in SRT tasks. For instance Gureckis & Love (2010, Cog Sci) observed an ``anti-recency effect'' in an SRT task when stimuli were less likely than chance to repeat at short lags. Perhaps the recency effect in this dataset reflects some kind of learning about the statistical structure rather than simply decaying activations. Additional control analyses could possibly tease this out from the existing data. Perhaps a recency bias reflects an prior about the statistics of the world (Anderson & Schooler, 1991, Psych Sci; Gershman, et al., 2014, PLoS Comp Bio)?

The other interesting result is that, by comparing RTs on specific motifs, the results favor a model in which the SR is constructed by updating with the current memory trace rather than bootstrapping via temporal difference learning. So, what to make of all this in the context of existing modeling on SR? Well, if the eligibility trace is allowed to cue subsequent states in addition to the SR, this would account for the recency effect. In models of laboratory memory, the exponentially-decaying eligibility trace is sometimes referred to as a temporal context vector (Gershman, ... Sederberg, 2012, Neural Comp). And the SR constructed from trace update rather than bootstrapping (and with \gamma=0) is (more or less) the temporal context model. Is this account falsified by the data? Presumably one would need to disentangle the exponential decay over powers of the transition matrix from \gamma and the exponential decay due to \lambda. In the temporal community graph at least, these seem to be perfectly confounded with one another, no?

The paper would be strengthened a great deal by making the connection between the recency+trace SR model and temporal context models. This class of models has been extensively studied in laboratory memory tasks (although not any SRT papers that come to mind).

Expository point:

Methods/results should be fleshed out and, at least high level methods should be embedded. (If there's a supplement/appendix with this information, it didn't make it to this reviewer.) As it is, there are a bunch of results missing that make it difficult to evaluate the paper's empirical contribution and what can really be learned from the modeling. I don't see values of \beta_trial, \beta_{ntrials}, \beta_10 reported in the paper. One can learn a lot about whether the estimates of these parameters change across models (they should not). More broadly, the data should be reported in more detail. How do distributions of parameters across participants look? Do parameters trade off with one another?

Minor Methodological points:

There are some claims/suggestions that could be explicitly tested to be on solid empirical ground. For instance, ``Interestingly on the average over participants, we estimate a remark- ably consistent degree of temporal discounting across all three graph topologies..'' To make this observation one should test the model that supports this inference. The \gamma horizon seems to be about as big as could be sustained with these graphs, no? To seriously test the hypothesis, perhaps one can fix \gamma to be the same for all three graphs and compare that to the (nested) model with the \gamma's allowed to vary freely?

All of the analyses in the paper model the effect of various variables on mean log RT. This is not unreasonable, but relies on an implicit assumption about the form of the RT distributions. Mean log RT is a pretty good proxy for drift rate in an evidence accumulation framework. But in at least some circumstances expectation is carried by non-decision time (Tiganj et al., 2022, JEP:G). And in at least some circumstances, recency is carried by the non-decision time (Bright, et al., 2022, bioRxiv). A change in non-decision time across conditions would not only change mean log RT but also \sigma. Tracking this down may turn out to be intricate but one could ask simple questions in an exploratory way. For instance, examination of RT distributions as a function of recency, or within/across-cluster transitions (etc) could be informative. Another straightforward approach would be to explicitly allow sigma as well as mu to vary for different variables. Those models are nested. As an aside, the shape of the RT distributions associated with the entropy effect (from the 2020 Nat Rev Phys paper) are also very interesting.

Expository point related to the neural understanding of SR:

The discussion writes: ``Alternatively, neurons representing eligible states might be reactivated after a delay, via history stored in some separate memory system '' True, but there is also very strong evidence for slowly changing neural activation expressed in firing rates. There is pretty good evidence for exponential decay of firing, as predicted by eligibility trace representations (e.g., Danskin et al., 2023, Sci Adv), in a variety of regions in a variety of tasks. However, unlike the standard RL account there appears to be a continuous spectrum of time constants across neurons within the population. Wouldn't Hebbian learning with a multiscale eligibility trace give a multi-scale SR?

**Have the authors made all data and (if applicable) computational code underlying the findings in their manuscript fully available?**

Reviewer #1: Yes

Reviewer #2: None

PLOS authors have the option to publish the peer review history of their article (what does this mean?). If published, this will include your full peer review and any attached files.

Reviewer #1: **Yes: **Sven Wientjes

Reviewer #2: **Yes: **Marc William Howard

**Figure resubmission:**
---

## [Decision Letter · Decision Letter 1]

3 Nov 2025

Dear Dr. Kahn,

We are pleased to inform you that your manuscript 'Trial-by-trial learning of successor representations in human behavior' has been provisionally accepted for publication in PLOS Computational Biology.

Please also use this opportunity to fix the final issue (the possible deletion of the new limitation paragraph, see below).

Best regards,

Wolfgang Einhäuser

Academic Editor

PLOS Computational Biology

Lyle Graham

Section Editor

PLOS Computational Biology

I agree with reviewer #2 that the new limitations section is not really necessary, but to avoid an additional iteration, I would ask you to delete it (if you agree) when submitting the production material.

Reviewer's Responses to Questions

**Comments to the Authors:**

Reviewer #1: The authors have satisfactorily addressed all of my previous comments. Appropriate parameter and model recovery analyses are now reported. I am very excited to see the authors tested the reliability of the parameter estimates to the empirical data, and understand the reasoning to not include this in the manuscript.

The expanded explanation of the roles of lambda and gamma is very helpful. As someone only superficially familiar with eligibility traces, I previously viewed them mainly as a computational shortcut to speed up learning. The revised text clarified that the inclusion of gamma in the decay also affects the final convergence of the model. I think this point could still be stated a bit more clearly, but the current version already conveys the essential idea well.

Overall, the manuscript now makes a strong and valuable contribution to the literature on computational models of sequence learning and to the growing dialogue between reinforcement learning frameworks and models of episodic memory.

Reviewer #2: The revision has addressed my concerns.

I am happy with the response to my point 6---the RT distributions are pretty intriguing---but don't think the text on page 18 discussing it (starting with ``Limitations'') adds much. I agree that this is a topic for future investigation and would recommend omitting that paragraph.

**Have the authors made all data and (if applicable) computational code underlying the findings in their manuscript fully available?**

Reviewer #1: Yes

Reviewer #2: None

PLOS authors have the option to publish the peer review history of their article (what does this mean?). If published, this will include your full peer review and any attached files.

Reviewer #1: **Yes: **Sven Wientjes

Reviewer #2: No

---

## [Editor Report · Acceptance letter]

PCOMPBIOL-D-25-00267R1

Trial-by-trial learning of successor representations in human behavior

Dear Dr Kahn,

I am pleased to inform you that your manuscript has been formally accepted for publication in PLOS Computational Biology. Your manuscript is now with our production department and you will be notified of the publication date in due course.

With kind regards,

Anita Estes
